# Estimating individual treatment effect on disability progression in multiple sclerosis using deep learning

Jean-Pierre R. Falet [1,2,3] ✉, Joshua Durso-Finley[2,3], Brennan Nichyporuk[2,3], Julien Schroeter[2,3], Francesca Bovis[4], Maria-Pia Sormani [4,5], Doina Precup[3,6], Tal Arbel[2,3] & Douglas Lorne Arnold [1,7]

Disability progression in multiple sclerosis remains resistant to treatment. The absence of a suitable biomarker to allow for phase 2 clinical trials presents a high barrier for drug development. We propose to enable short proof-of-concept trials by increasing statistical power using a deep-learning predictive enrichment strategy. Specifically, a multi-headed multilayer perceptron is used to estimate the conditional average treatment effect (CATE) using baseline clinical and imaging features, and patients predicted to be most responsive are preferentially randomized into a trial. Leveraging data from six randomized clinical trials ($n = 3,830$), we first pre-trained the model on the subset of relapsing-remitting MS patients ($n = 2,520$), then fine-tuned it on a subset of primary progressive MS (PPMS) patients ($n = 695$). In a separate held-out test set of PPMS patients randomized to anti-CD20 antibodies or placebo ($n = 297$), the average treatment effect was larger for the 50% (HR, 0.492; 95% CI, 0.266-0.912; $p = 0.0218$) and 30% (HR, 0.361; 95% CI, 0.165-0.79; $p = 0.008$) predicted to be most responsive, compared to 0.743 (95% CI, 0.482-1.15; $p = 0.179$) for the entire group. The same model could also identify responders to laquinimod in another held-out test set of PPMS patients ($n = 318$). Finally, we show that using this model for predictive enrichment results in important increases in power.

Several disease-modifying therapies have been developed for the treatment of the focal inflammatory manifestations of relapsing-remitting multiple sclerosis (RRMS) (clinical relapses and lesion activity) using the strategy of performing relatively short and small phase 2 trials with a magnetic resonance imaging (MRI) endpoint. These were meant to establish proof-of-concept and find the optimal dose, before proceeding to longer, more expensive phase 3 trials. In contrast to focal inflammatory manifestations, the absence of analogous MRI endpoints for disability progression independent of relapses has hampered progress in developing drugs for this aspect of the disease. Progressive biology predominates in progressive forms of multiple sclerosis, but is increasingly appreciated to be important in RRMS[1]. Although brain atrophy has been used as a biomarker of progression in phase 2 trials of progressive disease, its ability to predict the effect on disability progression in subsequent phase 3 clinical trials remains uncertain. As proceeding directly to large, phase 3 trials is expensive and risky, most programs that followed this path have failed to adequately demonstrate efficacy.

It is often the case that medications are more effective in some patients than others. Selecting such a subgroup for inclusion in a

[1]Montreal Neurological Institute, Department of Neurology and Neurosurgery, McGill University, Montreal, QC, Canada. [2]Centre for Intelligent Machines, Department of Electrical and Computer Engineering, McGill University, Montreal, QC, Canada. [3]Mila-Quebec AI Institute, Montreal, QC, Canada. [4]Department of Health Sciences (DISSAL), University of Genoa, Genoa, Italy. [5]IRCCS Ospedale Policlinico San Martino, Genoa, Italy. [6]School of Computer Science, McGill University, Montreal, QC, Canada. [7]NeuroRx Research, Montreal, QC, Canada. ✉e-mail: jean-pierre.falet@mail.mcgill.ca

clinical trial in order to increase its power is a technique called *predictive enrichment*[2]. A drug proven to be efficacious in a trial enriched with predicted responders can later be tested more confidently in a population predicted to be less responsive. This sequence prevents efficacious medications from having their effect diluted in early clinical trials due to the inclusion of a population that is too heterogeneous, while still allowing for broadening of indication criteria. It also improves the balance of risks and benefits for participants, since those who are unlikely to benefit from a drug would not be exposed to it and therefore would not experience potential adverse effects. A relevant application of predictive enrichment was described by Bovis et al.[3], who used Cox Proportional Hazard s (CPH) models to successfully predict a more responsive subgroup of RRMS patients to laquinimod, a medication whose average treatment effect in the original phase 3 studies was insufficient for drug approval.

Deep learning is a highly expressive and flexible type of machine learning that can potentially uncover complex, non-linear relationships between baseline patient characteristics and their responsiveness to treatment. However, contrary to traditional machine learning problems where a mapping between features and targets is learned from a sample of observations, the target in a treatment response (or treatment effect) task is not directly observable. Adaptations to machine learning frameworks must therefore be made in order to frame the problem through the lens of causal inference (reviewed in detail in the survey on uplift modeling by Gutierrez et al.[4]). Arguably some of the most popular methods have been tree-based approaches[5] which model treatment effect directly, and meta-learning approaches[6] which decompose the treatment effect estimation problem into simpler problems that can be tackled using traditional machine learning models. In a recent paper, Durso-Finley et al. used a meta-learning approach for the estimation of treatment effect (as measured by suppression of new/enlarging T2 lesions) in RRMS using baseline brain MRI and clinical variables[7].

In this work, we present a new deep learning framework to estimate an individual's treatment effect on disability progression using readily available clinical information (demographic characteristics and clinical disability scores) and scalar MRI metrics (lesional and volumetric) obtained at the screening visit of a clinical trial. This approach, based on an ensemble of multi-headed multilayer perceptrons (MLPs), can identify more responsive individuals to both anti-CD20 monoclonal antibodies (anti-CD20-Abs) and laquinimod better than alternative strategies. We demonstrate how using this model for predictive enrichment could greatly improve the feasibility of short proof-of-concept trials studying the effect of novel treatments for progression, thus accelerating therapeutic advances.

## Results
### Datasets
Data were pooled from six randomized clinical trials (*n* = 3830): OPERA I[8], OPERA II[8], BRAVO[9], ORATORIO[10], OLYMPUS[11], and ARPEGGIO[12] (ClinicalTrials.gov numbers, NCT01247324, NCT01412333, NCT00605215, NCT01194570, NCT00087529, NCT02284568, respectively). OPERA I/II, and BRAVO were RRMS trials which compared ocrelizumab with subcutaneous interferon beta-1a (IFNb-1a), and laquinimod with both intramuscular IFNb-1a and placebo, respectively. ORATORIO, OLYMPUS, and ARPEGGIO were placebo-controlled primary progressive multiple sclerosis (PPMS) trials which studied ocrelizumab, rituximab, and laquinimod, respectively.

The dataset is divided into three subsets for different phases of training and evaluation. The first subset (*n* = 2520) contains data from the three RRMS trials, and is used for pre-training the MLP to learn predictors of treatment effect under the RRMS condition (for details, see the section "Methods"). This pre-training phase falls under the umbrella of transfer learning, a deep learning strategy that is used to transfer knowledge acquired from a related task to a task with fewer

samples in order to improve learning on the latter[13]. Importantly, the RRMS dataset is only used for pre-training and does not take part in the final model evaluation, since this study is focused on the challenge of improving the efficiency of clinical trials for progressive MS. The second subset consists of two PPMS trials (*n* = 992): OLYMPUS and ORATORIO. This subset is divided into a 70% training set (*n* = 695) which is used to fine-tune the pre-trained MLP to estimate treatment effect on anti-CD20-Abs, and the remaining 30% (*n* = 297) is held out as a test set to estimate the generalization error of the fully trained model. The third subset contains PPMS data from the trial ARPEGGIO (*n* = 318), which is also held out as a second test set.

Mean and standard deviation for the baseline features and the outcome metrics in the PPMS subset are shown in Table 1, separated by treatment arm (the same statistics for the RRMS subset are shown in Supplementary Table 1). The groups are comparable for all features except for disease duration which is shorter in ORATORIO, and Gad count and T2 lesion volume, which are greater in ORATORIO. This may be due to ORATORIO's inclusion criteria, which had a maximum time from symptom onset, and to inter-trial differences in automatic lesion segmentation, which are accounting for using a scaling procedure explained in the section "Data". Some heterogeneity exists between the outcomes of each trial when looking at the placebo arms, which on average have a smaller restricted mean survival time (RMST) at 2 years in ARPEGGIO and OLYMPUS compared to ORATORIO, indicating more rapid disability progression on the Expanded Disability Status Scale (EDSS).

### Predicting response to anti-CD20 monoclonal antibodies
As described in the section "Methods", we train an ensemble of multi-headed MLPs to predict the change in EDSS over time (obtained by fitting a linear regression model to an individual's EDSS values recorded over time and taking the slope of the regression to be the prediction target) on both anti-CD20-Abs and placebo. These two predictions are then subtracted to obtain an estimate of the conditional average treatment effect (CATE) for each individual, given their baseline features. The CATE estimate is used to infer an individual's treatment effect, as explained in the section "Treatment effect modeling".

The fully trained model is then evaluated on the held-out anti-CD20-Abs test set (30% of the dataset; *n* = 297). A histogram of predictions on this test set is shown in Supplementary Fig. 1. The model's ability to rank response is assessed using an average difference curve, AD(*c*), which is described by Zhao et al.[14] and is well suited for measuring performance in predictive enrichment. Our implementation measures the ground-truth average difference in RMST (calculated at 2 years from time to 24-week confirmed disability progression (CDP24)) between anti-CD20-Abs and placebo for individuals predicted to respond more than a certain threshold, as a function of this threshold. The AD(*c*) curve for our model, shown in Fig. 1, appropriately increases as a subgroup that is predicted to be more and more responsive is selected. The $AD_{wabc}$, a metric derived from the area under the AD(*c*) curve in Supplementary Methods 3, provides a measure of how well the model can rank individuals on the basis of their responsiveness to treatment. Larger positive $AD_{wabc}$ values indicate better performance. The $AD_{wabc}$ in this case is positive, relatively large (0.0565), and nearly monotonic (Spearman r correlation coefficient 0.943), demonstrating the ability for the model to rank response to anti-CD20-Abs.

Kaplan-Meyer curves of the ground-truth time-to-CDP24 for predicted responders in the test set are shown in Fig. 2 for two predictive enrichment thresholds (selecting the 50% or the 30% that are predicted to be most responsive). The Kaplan–Meyer curves for corresponding non-responder groups (the 50% and 70% predicted to be least responsive) are also shown. Compared to the entire test set, whose HR is 0.743 (95% CI, 0.482–1.15; *p* = 0.179), predictive enrichment leads to a HR of 0.492 (95% CI, 0.266–0.912; *p* = 0.0218) and 0.361 (95% CI,

**Table 1 | Baseline features and outcomes per treatment arm**

| | Ocrelizumab | Rituximab | Laquinimod | Placebo | | |
| | ORATORIO<br>n = 436 | OLYMPUS<br>n = 212 | ARPEGGIO<br>n = 186 | ORATORIO<br>n = 225 | OLYMPUS<br>n = 119 | ARPEGGIO<br>n = 132 |
|---|---|---|---|---|---|---|
| *Demographics* | | | | | | |
| Age (years) | 44.50 (7.90) | 49.54 (9.01) | 46.35 (6.62) | 44.41 (8.40) | 49.89 (8.68) | 46.70 (7.16) |
| Sex (% male) | 51.61 | 48.11 | 56.45 | 47.56 | 43.70 | 50.76 |
| Height (cm) | 170.20 (9.61) | 170.77 (9.30) | 172.11 (9.41) | 170.20 (9.57) | 169.87 (8.90) | 171.23 (9.73) |
| Weight (kg) | 72.35 (17.26) | 78.13 (16.37) | 75.25 (15.40) | 72.51 (15.24) | 77.60 (17.13) | 73.20 (16.21) |
| Disease duration (years) | 6.56 (3.77) | 9.03 (6.25) | 8.12 (6.07) | 6.01 (3.38) | 8.59 (6.81) | 7.41 (5.23) |
| *Disability scores* | | | | | | |
| EDSS | 4.69 (1.18) | 4.79 (1.36) | 4.49 (0.98) | 4.65 (1.16) | 4.58 (1.41) | 4.46 (0.91) |
| FSS-Bowel and Bladder | 1.14 (0.85) | 1.42 (0.95) | 1.27 (0.95) | 1.14 (0.91) | 1.21 (0.94) | 1.16 (0.88) |
| FSS-Brainstem | 0.88 (0.91) | 0.75 (0.90) | 1.01 (0.92) | 0.89 (0.93) | 0.61 (0.81) | 0.98 (0.95) |
| FSS-Cerebellar | 2.11 (0.98) | 2.03 (1.12) | 2.11 (0.83) | 2.14 (0.89) | 1.99 (1.10) | 2.10 (0.89) |
| FSS-Cerebral | 0.91 (0.88) | 1.30 (0.84) | 0.93 (0.91) | 0.91 (0.82) | 1.24 (0.89) | 0.86 (0.88) |
| FSS-Pyramidal | 2.87 (0.62) | 2.69 (0.82) | 2.92 (0.55) | 2.83 (0.65) | 2.82 (0.78) | 2.85 (0.66) |
| FSS-Sensory | 1.58 (1.04) | 1.48 (0.99) | 1.73 (1.04) | 1.53 (1.07) | 1.52 (1.11) | 1.74 (1.01) |
| FSS-Visual | 0.79 (0.87) | 0.86 (1.04) | 0.92 (1.30) | 0.71 (0.82) | 0.91 (1.05) | 0.79 (1.10) |
| Mean T25FW (s) | 13.93 (18.44) | 11.74 (14.56) | 9.61 (8.85) | 11.71 (12.35) | 11.01 (13.65) | 9.68 (7.54) |
| Mean 9HPT dominant (s) | 34.09 (33.99) | 28.80 (17.60) | 28.57 (12.37) | 31.67 (21.50) | 27.22 (10.22) | 28.22 (12.15) |
| Mean 9HPT non-dominant (s) | 36.05 (38.50) | 31.88 (24.99) | 31.44 (18.04) | 37.51 (40.29) | 30.95 (17.50) | 29.04 (12.16) |
| *MRI metrics* | | | | | | |
| Gad count | 1.23 (5.36) | 0.63 (2.47) | 0.27 (0.81) | 0.56 (1.47) | 0.47 (1.14) | 0.45 (1.84) |
| T2 lesion volume (mL) | 12.45 (14.92) | 8.44 (10.50) | 5.86 (9.11) | 11.33 (13.27) | 8.57 (11.66) | 5.96 (8.65) |
| Normalized brain volume (L) | 1.46 (0.08) | 1.20 (0.12) | 1.46 (0.10) | 1.47 (0.09) | 1.21 (0.12) | 1.46 (0.11) |
| *Outcome* | | | | | | |
| Slope (EDSS change/yr)[a] | 0.22 (0.53) | 0.27 (0.65) | 0.32 (0.77) | 0.27 (0.71) | 0.39 (0.63) | 0.28 (0.64) |
| RMST (at 2 years)[b] | 1.92 | 1.89 | 1.69 | 1.91 | 1.87 | 1.72 |

Values in brackets are standard deviations, unless otherwise specified.

[a] Slope is based on the coefficient of regression from a linear regression model that is fit on an individual's EDSS values over time, as described in the section "Outcome definition".

[b] RMST calculated at 2 years using time to 24-week confirmed disability progression on the EDSS.

*RMST* restricted mean survival time, *EDSS* Expanded Disability Status Scale, *FSS* Functional Systems Score, *T25FW* timed 25-foot walk, *9HPT* 9-hole peg test, *Gad* gadolinium-enhancing lesion.

0.165–0.79; $p = 0.008$) when selecting the 50% and 30% most responsive, respectively. The corresponding non-responder groups have a HR of 1.11 (95% CI, 0.599–2.05; $p = 0.744$) and 0.976 (95% CI, 0.578–1.65; $p = 0.925$) when selecting the 50% and 70% least responsive, respectively. This heterogeneity suggests that a significant part of the trend for effect at the whole-group level may be explained by a small proportion of more responsive patients.

Of ocrelizumab and rituximab, only the former had a significant effect in a phase 3 trial (ORATORIO), and it is the only drug approved in PPMS. We therefore verified whether the model's enrichment capabilities are maintained within the ORATORIO subgroup ($n = 188$) of the test set, which has HR of 0.661 (95% CI 0.383–1.14, $p = 0.135$). If selecting the 50% ($n = 96$) and 30% ($n = 57$) predicted to be most responsive, the HR reduces to 0.516 (95% CI, 0.241–1.1; $p = 0.084$) and 0.282 (95% CI, 0.105–0.762; $p = 0.0082$), respectively. The corresponding 50% and 70% predicted to be least responsive have a HR of 0.849 (95% CI, 0.385–1.87; $p = 0.685$) and 0.915 (95% CI, 0.471–1.78; $p = 0.791$), respectively.

We then considered specific demographic subgroups to understand their effect on model performance. For men, the model achieved a $AD_{wabc}$ of 0.0405, while for women the model performs better ($AD_{wabc} = 0.0844$). For those with an age <51, the $AD_{wabc}$ of 0.0353 is lower than for those with an age ≥51 ($AD_{wabc} = 0.0661$). For those with a disease duration < 5, the model performs less well than on those with a disease duration ≥5 ($AD_{wabc} = 0.0385$ compared to 0.0117). Finally, the

model performs better for those with an EDSS <4.5 ($AD_{wabc} = 0.069$) than for those with an EDSS of ≥4.5 ($AD_{wabc} = 0.0451$).

Group characteristics for the predicted responders and non-responders, defined at the 50th and 70th percentile thresholds, are shown in Table 2. We observe enrichment across a broad range of input features in the responder subgroups: younger age, shorter disease duration, higher disability scores, and more lesional activity (particularly T2 lesion volume). The largest effect on the Functional Systems Scores (FSS) was seen in Cerebellar and Visual sub-scores, while FSS-Bowel and Bladder, Brainstem, Cerebral, Pyramidal, and Sensory did not reach statistical significance ($p < 0.05$). Timed 25-foot walk (T25FW) was significantly different only for the 70th percentile threshold. Normalized brain volume was the only baseline MRI feature which did not differ significantly between the two groups at either threshold.

**Predicting response to laquinimod**

To determine whether the same model trained on the anti-CD20-Abs dataset could be predictive of treatment response to a medication with a different mechanism of action, and to provide a second validation for the model trained on the single 70% training set in the first anti-CD20-Abs experiment, we tested it on data from ARPEGGIO ($n = 318$). The model trained on the anti-CD20-Abs training dataset also generalized to this second test set, as shown by a positive $AD_{wabc} = 0.0211$. From the whole-group HR of 0.667 (95% CI: 0.369–1.2; $p = 0.933$), selecting the

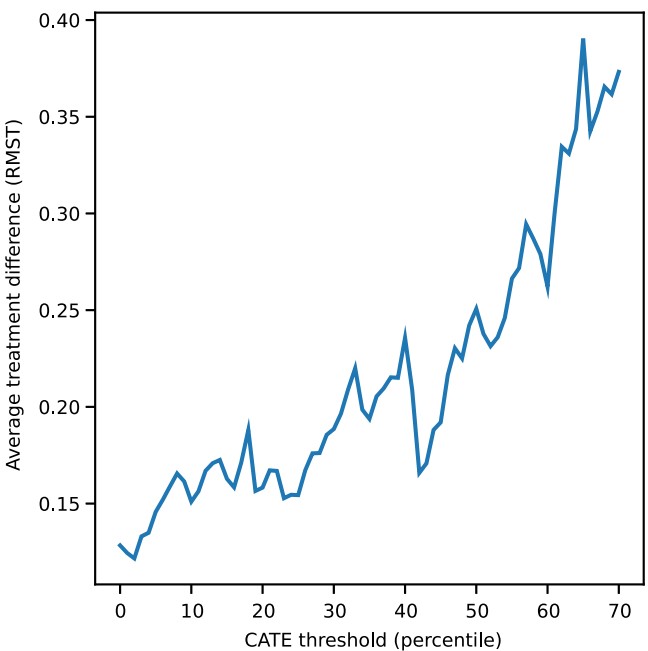

**Fig. 1 | Average treatment difference curve for the anti-CD20-Abs held-out test set.** Represents the difference in the ground-truth restricted mean survival time (RMST), calculated at 2 years using time-to-CDP24, between anti-CD20-Abs and placebo, among predicted responders defined using various thresholds. The conditional average treatment effect (CATE) percentile threshold is the minimum CATE (expressed as a percentile among all CATE estimates in the test set) that is used to define an individual as a responder (i.e. a threshold of 0.7 means the 30% predicted to be most responsive are considered responders).

50% and the 30% predicted to be most responsive yields a HR of 0.492 (95% CI 0.219–1.11; $p = 0.0803$) and 0.338 (95% CI, 0.131–0.872; $p = 0.0186$), respectively. The corresponding 50% and 70% predicted to be least responsive have a HR of 0.945 (95% CI, 0.392–2.28; $p = 0.901$) and 0.967 (95%CI, 0.447–2.09; $p = 0.933$), respectively. The Kaplan–Meyer curves for these predicted subgroups are shown in Supplementary Fig. 2.

Group characteristics for predicted responders are shown in Supplementary Table 2. Groupwise differences are largely similar to those obtained on the anti-CD20-Abs dataset, with a few exceptions. In the laquinimod dataset, a significantly greater FSS-Bowel and Bladder and smaller normalized brain volume (NBV) are observed (whereas these did not reach the same level of significance in the anti-CD20-Abs test set), and the difference in T25FW is not statistically significant ($p < 0.05$). A smaller NBV was found in the responder group, but this only reached significance at the 50th percentile threshold. Nonetheless, the direction of the effect for these differences is concordant between the two test sets.

### Comparison to baseline models
The performance of the non-linear model described in this paper is compared to numerous other baseline models in Table 3, as measured by the $AD_{wabc}$ on the anti-CD20-Abs test set and on the laquinimod dataset. Scatter plots of the metrics obtained on both test sets are also provided in Supplementary Fig. 3, 4. All models were trained using the same procedure, on the same dataset, and with the same regression target. The MLP outperforms all other baselines on this metric, but some models (such as a linear regression model with L2 regularization (ridge regression) and a CPH model) compare favorably on one of the two datasets. Without pre-training on the RRMS dataset, the performance of the MLP is still strong but inferior to the fine-tuned model. All single feature models are inferior to the MLP and CPH models except

for the T2 lesion volume/disease duration model which falls between these two models in terms of performance on the anti-CD20-Abs test set. We also tested a prognostic MLP which is only trained to predict progression on placebo, and which uses this prediction in place of the CATE estimate (assumes that more rapid progression leads to greater potential for treatment effect). This model's performance on the anti-CD20-Abs test set falls between that of the CPH model and the T2 lesion volume/disease duration model.

In OLYMPUS, Hawker et al.[11] identify a cutoff of age <51 years and gadoliniumenhancing (Gad) lesion count >0 at baseline as predictive of treatment effect. Using their definition, 21.9% and 11.3% of the patients in the anti-CD20-Abs and laquinimod datasets, respectively, would be classified as responders. This is more restrictive than our most restrictive threshold which selects the 30% predicted to be most responsive. The HR for these predicted responders is 0.91 (95% CI, 0.392–2.11; $p = 0.831$) and 0.305 (95% CI, 0.0558–1.67; $p = 0.147$) for the anti-CD20-Abs and the ARPEGGIO patients, respectively. For both datasets, these effect size estimates do not reach statistical significance ($p < 0.05$). The effect size estimate for the anti-CD20-Abs dataset is also smaller compared to that obtained with our predictive enrichment method when selecting the 30% most responsive individuals. This binary cutoff is therefore generally inferior to our approach.

Finally, we compared our approach to the traditional phase 2 approach which typically uses an MRI-based surrogate outcome (brain atrophy being the most common) which is thought to be correlated with the clinical outcome of interest but that is more sensitive to the underlying biological processes or that has a lower variance, in order to increase a study's statistical power. For example, suppose our anti-CD20-Abs test set ($n = 297$) was a small phase 2 trial testing anti-CD20-Abs with brain atrophy as the primary outcome. Measuring brain atrophy at the 48-week MRI for the anti-CD20-Abs, the mean difference between the treatment arms is 0.066 (95% CI, −0.397 to 0.529; $p = 0.7786$). Looking at ORATORIO patents separately, since ORATORIO was the only positive trial in the anti-CD20-Abs dataset, the mean difference is 0.110 (95% CI, −0.352 to 0.572; $p = 0.6379$). Brain atrophy would therefore not have been able to detect a significant effect for ocrelizumab or for anti-CD20-Abs.

### Simulating a phase 2 clinical trial enriched with predicted responders
To understand the effect of enriching a future clinical trial studying novel B-cell depleting agents, we simulated both a one and a two-year randomized clinical trial using populations enriched with predicted responders and estimated the sample size that would be needed to detect a significant effect under these conditions. To do so, we first used our model to predict responders to anti-CD20-Abs, defined using a range of thresholds (from including all individuals to including only the top 30% who are predicted to be most responsive). We then fit a CPH model to the ground-truth time-to-CDP24 for the responder group obtained at each threshold in order to estimate their corresponding HR. We then used the observed one-year and two-year CDP24 event rates in each responder group to calculate the sample size needed to detect a significant effect during a one-year or a two-year trial, respectively. This analysis is shown in Table 4.

Using the 50th percentile as a threshold for randomization in a two-year long trial as an example, a total of 490 individuals would be screened and the top 50% who are predicted to be most responsive would be randomized ($n = 245$). This leads to a six-fold reduction in the number of patients that need to be randomized while screening almost three times less patients compared to the scenario where all participants are randomized into a two-year study ($n = 1374$).

### Discussion
This work addresses the lack of a sufficiently predictive biomarker of treatment response for progression in multiple sclerosis, which has

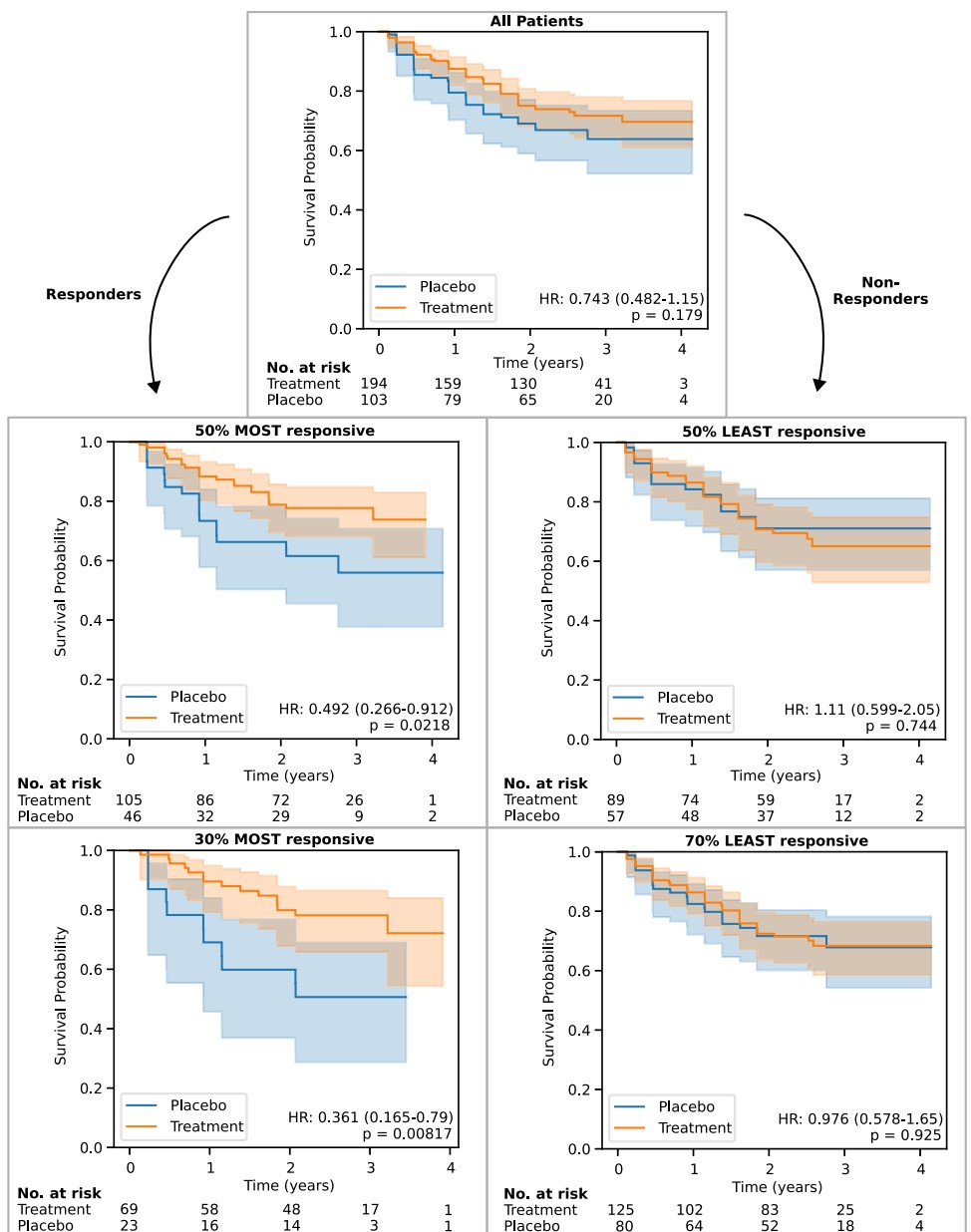

**Fig. 2 | Kaplan–Meyer curves ±95% confidence intervals (CI) for predicted responders and non-responders to anti-CD20-Abs in the held-out test set, defined at two thresholds of predicted effect size.** These are compared to the whole group (top). The placebo group is displayed in blue, and the treatment (anti-CD20-Abs) group is displayed in orange. Survival probability is measured in terms of time-to-CDP24 using the EDSS. *p* values are calculated using log-rank tests. 95% CIs are estimated using Greenwood's Exponential formula.

hampered progress by preventing efficient phase 2 clinical trials. We describe a deep learning solution to increasing the efficiency of early proof-of-concept clinical trials based on a multi-headed MLP architecture designed for CATE estimation. This approach can consistently identify and rank treatment effect among patients exposed to anti-CD20-Abs, and could reduce by several fold the sample size required to detect an effect in a short one or two-year long trial. We validate our model using a dataset composed of patients exposed to anti-CD20-Abs, and a second dataset of patients exposed to laquinimod. We demonstrate that a model trained to predict response to anti-CD20-Abs can also generalize to laquinimod, a medication with a very different mechanism of action, suggesting that there exists disease-agnostic predictors of response.

The model's predicted responders were enriched in numerous baseline features, including a younger age, shorter time from symptom onset, higher disability scores, and more lesion activity. Similarly, in

subgroup analyses from OLYMPUS, an age less than 51 years and the presence of Gad lesions at baseline was also found to be associated with increased response[11]. Signori et al.[15] also found that younger age and the presence of Gad were associated with greater treatment effect in RRMS. In a study by Bovis et al.[3], a response scoring function obtained via CPH models in RRMS also identified Gad lesions and a higher normalized brain volume as predictive of treatment effect, although older age was found to be more predictive in the combination they studied.

In our experiments, a non-linear model (MLP) outperformed other linear (and log-linear) baselines, suggesting that complex relationships exist between the baseline features and treatment effect. Nonetheless, a prognostic model (that predicts response to a medication solely based on the prediction of progression on placebo) also performed well, suggesting that poor prognosis is also predictive of treatment effect. A prognostic model could therefore

**Table 2 | Group statistics for predicted responders and non-responders to anti-CD20-Abs at the 50th and 70th percentile thresholds, in the held-out test set**

| | 50th percentile threshold[a] | | | | 70th percentile threshold[a] | | | |
|---|---|---|---|---|---|---|---|---|
| | Responders | Non-responders | Effect size (95% CI)[b] | p value[c] | Responders | Non-responders | Effect size (95% CI)[b] | p value[c] |
| *Trial contribution* | | | | | | | | |
| OLYMPUS | 55 | 54 | | | 35 | 74 | | |
| ORATORIO | 96 | 92 | | | 57 | 131 | | |
| *Demographics* | | | | | | | | |
| Age (years) | 45.20 (8.58) | 47.84 (7.89) | −2.64 (−4.53, −0.76) | 0.006 | 44.59 (9.05) | 47.36 (7.87) | −2.77 (−4.93, −0.61) | 0.013 |
| Sex (% male) | 47.02 | 50.68 | 0.86 (0.53, 1.40) | 0.562 | 45.65 | 50.24 | 0.83 (0.49, 1.40) | 0.530 |
| Height (cm) | 170.05 (10.56) | 170.55 (8.80) | −0.50 (−2.72, 1.71) | 0.657 | 169.78 (10.29) | 170.52 (9.47) | −0.74 (−3.23, 1.75) | 0.560 |
| Weight (kg) | 76.17 (18.93) | 72.96 (13.77) | 3.21 (−0.56, 6.98) | 0.096 | 75.68 (20.07) | 74.10 (14.87) | 1.58 (−3.04, 6.20) | 0.502 |
| Disease duration (years) | 6.07 (4.14) | 8.72 (5.45) | −2.65 (−3.76, −1.54) | <0.001 | 5.79 (4.15) | 8.09 (5.19) | −2.30 (−3.41, −1.19) | <0.001 |
| *Disability scores* | | | | | | | | |
| EDSS | 4.87 (1.18) | 4.52 (1.23) | 0.34 (0.07, 0.62) | 0.015 | 5.07 (1.14) | 4.53 (1.21) | 0.54 (0.25, 0.83) | <0.001 |
| FSS-Bowel and Bladder | 1.25 (0.93) | 1.11 (0.80) | 0.14 (−0.05, 0.34) | 0.157 | 1.27 (0.98) | 1.15 (0.82) | 0.12 (−0.11, 0.35) | 0.315 |
| FSS-Brainstem | 0.82 (0.93) | 0.79 (0.87) | 0.04 (−0.17, 0.24) | 0.726 | 0.90 (0.95) | 0.77 (0.88) | 0.13 (−0.10, 0.36) | 0.265 |
| FSS-Cerebellar | 2.38 (0.97) | 1.78 (1.05) | 0.60 (0.37, 0.83) | <0.001 | 2.57 (0.81) | 1.86 (1.08) | 0.71 (0.48, 0.93) | <0.001 |
| FSS-Cerebral | 1.07 (0.83) | 1.05 (0.89) | 0.02 (−0.18, 0.22) | 0.848 | 1.13 (0.84) | 1.04 (0.87) | 0.09 (−0.12, 0.30) | 0.404 |
| FSS-Pyramidal | 2.75 (0.69) | 2.90 (0.58) | −0.14 (−0.29, 0.00) | 0.052 | 2.77 (0.76) | 2.85 (0.58) | −0.08 (−0.26, 0.10) | 0.382 |
| FSS-Sensory | 1.55 (1.06) | 1.64 (1.02) | −0.08 (−0.32, 0.15) | 0.488 | 1.56 (1.00) | 1.61 (1.06) | −0.05 (−0.30, 0.20) | 0.703 |
| FSS-Visual | 1.04 (1.04) | 0.43 (0.62) | 0.62 (0.42, 0.81) | <0.001 | 1.28 (1.07) | 0.50 (0.71) | 0.78 (0.54, 1.02) | <0.001 |
| Mean T25FW (s) | 13.55 (17.61) | 10.75 (11.08) | 2.80 (−0.55, 6.15) | 0.103 | 15.95 (21.79) | 10.48 (9.82) | 5.47 (0.77, 10.17) | 0.024 |
| Mean 9HPT dominant (s) | 32.62 (26.89) | 26.70 (10.24) | 5.92 (1.29, 10.55) | 0.013 | 36.01 (33.25) | 26.88 (9.89) | 9.13 (2.12, 16.15) | 0.012 |
| Mean 9HPT non-dominant (s) | 37.33 (31.11) | 26.97 (9.32) | 10.36 (5.14, 15.58) | <0.001 | 42.39 (38.33) | 27.68 (9.33) | 14.71 (6.68, 22.75) | <0.001 |
| *MRI metrics* | | | | | | | | |
| Gad count | 1.62 (3.14) | 0.16 (0.48) | 1.46 (0.95, 1.97) | <0.001 | 1.90 (3.64) | 0.46 (1.27) | 1.44 (0.67, 2.22) | <0.001 |
| T2 lesion volume (mL) | 13.09 (12.85) | 7.72 (10.17) | 5.37 (2.73, 8.01) | <0.001 | 14.31 (14.22) | 8.72 (10.27) | 5.59 (2.33, 8.85) | <0.001 |
| Normalized brain volume (L) | 1.37 (0.16) | 1.38 (0.16) | −0.02 (−0.05, 0.02) | 0.367 | 1.35 (0.16) | 1.38 (0.16) | −0.03 (−0.07, 0.01) | 0.107 |

Values in brackets are standard deviations, unless otherwise specified.

*EDSS* Expanded Disability Status Scale, *FSS* Functional Systems Score, *T25FW* timed 25-foot walk, *9HPT* 9-hole peg test, *Gad* Gadolinium-enhancing lesion.

[a] Percentile threshold for defining responders. The 50th percentile defines responders as the top 50% who are predicted to be most responsive, while the 70th percentile defines them as the top 30%. The non-responders are those who fall below the percentile threshold.

[b] Effect size is the average difference between responders and non-responders for all covariates except for "sex" which is an odd's ratio (OR).

[c] p values for continuous and ordinal variables are calculated using a two-sided Welch's t test due to unequal variances/sample sizes. p value for the categorical variable "sex" is calculated using a two-sided Fisher's exact test due to unequal and relatively small sample sizes. Exact p-values for the 50th percentile threshold: disease duration, $p = 4.39 \times 10^{-6}$; FSS-Cerebellar, $p = 6.42 \times 10^{-7}$; FSS-Visual, $p = 2.18 \times 10^{-9}$; Mean 9HPT non-dominant, $p = 1.36 \times 10^{-4}$; Gad count, $p = 8.72 \times 10^{-8}$; T2 lesion volume, $p = 8.57 \times 10^{-5}$. Exact p-values for the 70th percentile threshold: disease duration, $p = 7.04 \times 10^{-5}$; EDSS, $p = 3.03 \times 10^{-4}$; FSS-Cerebellar, $p = 2.61 \times 10^{-9}$; FSS-Visual, $p = 3.38 \times 10^{-9}$; Mean 9HPT non-dominant, $p = 4.82 \times 10^{-4}$; Gad count, $p = 3.69 \times 10^{-4}$; T2 lesion volume, $p = 9.59 \times 10^{-4}$.

be helpful in cases where drugs with very different mechanisms of action (e.g., targeting remyelination, or neurodegeneration) are being tested, in which case a model trained to predict treatment effect on an anti-inflammatory drug might perform less well than a prognostic model.

Interestingly, despite a balanced dataset with respect to gender, our model was better at identifying responders in women compared to men. We also noted that the model performed better in individuals ≥51, disease duration <5 years, and/or an EDSS <4.5. These findings suggest further studies are needed to determine whether and why predictors of response might differ depending on the stage of disease and sex.

Predictive enrichment is not the only approach to increase the efficiency of clinical trials in PPMS. However, the traditional approach of using a potential surrogate marker (in this case brain atrophy) as part of a phase 2 study did not succeed in identifying a significant effect in our experiments, and may therefore limit early identification of effective therapies. Although brain atrophy has been used in phase 2 trials as a primary outcome, several studies on PPMS[16], RRMS[17], and secondary progressive multiple sclerosis (SPMS)[18] suggest no to

modest correlation with clinical disability progression based on EDSS even after four to eight years of follow-up.

Another strategy could have been to infer from an RRMS trial that a drug might be effective for treating disability progression in a progressive multiple sclerosis (PMS) trial. For example, ocrelizumab and siponimod were first found to be efficacious in the RRMS population in OPERA I/II[8] and BOLD[19], respectively, before being tested in the PPMS trial ORATORIO[10] and the SPMS trial EXPAND[20], respectively. In these cases, there were other reasons to believe that the drugs might be effective for treating progressive biology, but the predictive value of finding an initial effect on inflammatory biology remains of interest. From a predictive enrichment standpoint, baseline T2 lesion burden has been found to correlate with future disability and disability progression, at least modestly[21–24]. Evidence is less robust for Gad lesions, since some authors[25] have demonstrated modest correlations with future disability at least 2 years from baseline, while others[23] have not. In our experiments, a treatment effect estimation model based on either Gad count or T2 lesion volume alone performed poorly. Only the rate of accumulation of T2 lesions over time (measured from the time

**Table 3 | Comparison of model performance (measured by AD$_{wabc}$) on the held-out test set of patients from ORATORIO and OLYMPUS (anti-CD20-Abs), and on the held-out dataset from ARPEGGIO (laquinimod)**

| | Anti-CD20-Abs | Laquinimod |
|---|---|---|
| *Single feature*[a] | | |
| Negative disease duration | 0.0225 | 0.0114 |
| Negative age | 0.0067 | −0.0287 |
| Negative EDSS | 0.0264 | 0.0074 |
| Negative 9HPT dominant hand | −0.0109 | 0.0023 |
| Negative 9HPT non-dominant hand | −0.0012 | −0.0006 |
| Negative T25FW | 0.0033 | 0.0020 |
| T2 lesion volume | 0.0167 | −0.0051 |
| Gad count | 0.0021 | NaN[c] |
| *Feature/disease duration ratio*[b] | | |
| Age/disease duration | 0.0268 | 0.0138 |
| EDSS/disease duration | 0.0021 | 0.0020 |
| 9HPT dominant hand/disease duration | 0.0238 | 0.0146 |
| 9HPT non-dominant hand/disease duration | 0.0179 | 0.0098 |
| T25FW/disease duration | 0.0257 | 0.0049 |
| T2 lesion volume/disease duration | 0.0432 | 0.0164 |
| Gad count/disease duration | 0.0030 | NaN[c] |
| *Regression model using all features* | | |
| MLP (our model) | 0.0565 | 0.0211 |
| MLP (no pre-training[d]) | 0.0486 | 0.019 |
| MLP (prognostic model[e]) | 0.0408 | 0.0170 |
| Ridge Regression | 0.0227 | 0.0194 |
| *Survival model using all features* | | |
| CPH | 0.0305 | 0.0031 |

*EDSS* Expanded Disability Status Scale, *FSS* Functional Systems Score, *T25FW* timed 25-foot walk, *9HPT* 9-hole peg test, *Gad* Gadolinium-enhancing lesion, *MLP* Multi-layer perceptron.

[a] The value of the feature is taken to be the CATE estimate for an individual. For example, the "T2 lesion volume" model uses the value of an individual's T2 lesion volume as the CATE estimate for that individual, such that a larger baseline volume predicts a larger treatment effect. A "negative" feature implies that the CATE estimate is the negative of the value of the feature. For example, the "negative disease duration" model predicts a larger treatment effect with shorter disease duration.

[b] The value of the feature divided by the disease duration is taken to be the CATE estimate for an individual. For example, the "EDSS/disease duration" model predicts a larger treatment effect with a more rapid historical rate of change in the EDSS over time.

[c] Value for AD$_{wabc}$ could not be computed due to low variance in values for Gad lesions in the laquinimod dataset.

[d] This MLP was trained without pre-training on the RRMS dataset.

[e] The value of the predicted slope of disability progression on the placebo arm is used as the CATE estimate. In other words, a patient predicted to progress more rapidly on placebo (worse prognosis) predicts a larger treatment effect.

of symptom onset) was predictive. Even if the inflammatory hypothesis was correct, a predictive enrichment strategy would be more efficient than awaiting the results of a RRMS study testing the same drug, particularly given that the power of a follow-up PMS study is likely to be insufficient. This is supported by the small proportion of responders to anti-CD20-Abs in our experiments, the dramatic difference in effect size between the inflammatory and progression-related outcomes, and the numerous examples of effective drugs for RRMS that had no identifiable effect on slowing disability progression in PMS[11, 12, 26–30].

Finally, the Food and Drug Administration has published a guidance document with suggestions regarding the design of predictively enriched studies[31]. One approach might be to first conduct a small trial of a short duration as a proof of concept in patients predicted to be highly responsive. If a significant effect is detected, a larger/longer follow-up study with a more inclusive (less enriched) population can be attempted with more confidence. It is also possible that, on the basis of a strong effect in the enriched responder group, the proof of concept would be sufficient for drug approval to be granted for the unenriched population, given the significant unmet need and irreversible consequences of disability progression. To limit the risk that the predictive model is found to be inaccurate on the study population, stratified randomization can be used by having two parallel groups: the primary group (which would be adequately powered to detect an effect) would be an enriched responder group, while the secondary group would randomize predicted non-responders. Although the non-responder group would not be powered to detect an effect, it would provide a rough estimate of the effectiveness of the drug in this group and help guide design decisions for follow-up trials. The two groups could also be merged in a pre-planned analysis, to provide an estimate of the effect in the combined population.

Limitations of this work include the choice of model. Interpretability of black-box algorithms such as neural networks (reviewed elsewhere[32]) remains an area of active research. Although our MLP outperformed linear baselines, MLPs are more difficult to train and at higher risk of overfitting. Moreover, we made heavy use of several regularization schemes to prevent this. Our hyperparameter tuning procedure is also one of many that can be designed. Next, we used MRI-derived lesion and volumetric measures computed during the individual clinical trials, which could potentially ignore more subtle predictive features found within the MRI voxel-level data. Learning these features in a data-driven fashion through convolutional neural networks is the subject of ongoing work, but this can easily be appended to our MLP architecture. Regarding generalization to novel drug targets, more data is needed from drugs with diverse mechanisms of action to fully grasp the extent to which predictors of anti-inflammatory drugs are applicable to other drug classes, including neurodegenerative targets. Finally, it remains unknown if patients for whom our model predicted minimal effect over two to four years could benefit after longer periods of administration. Answering this question would require longer-term observational data.

## Methods

The study protocol was originally approved by the McGill University Health Center's Research Ethics Board - Neurosciences-Psychiatry (IRB00010120) and then transferred and approved by the McGill University Faculty of Medicine and Health Sciences Institutional Review Board (A03-M14-22A).

### Data

Data is taken from six different randomized clinical trials (*n* = 3830): OPERA I[8], OPERA II[8], BRAVO[9], ORATORIO[10], OLYMPUS[11], and ARPEGGIO[12] (ClinicalTrials.gov numbers, NCT01247324, NCT01412333, NCT00605215, NCT01194570, NCT00087529, NCT02284568, respectively). Informed consent and participant compensation (if any) were handled by the individual clinical trials. We excluded participants who spent less than 24 weeks in the trial, who had less than two clinical visits, or who were missing one or more input features at the baseline visit. Therefore, it is important to appreciate that the data included in our work are not an exact reproduction of those used in the clinical trials.

All clinical/demographic and MRI features that were consistently recorded as part of all 6 clinical trials (a total of 19 features) were used to train our model. Values were recorded at the baseline visit (immediately before randomized treatment allocation), and are a combination of binary (sex), ordinal (EDSS, FSS), discrete (Gad count), and continuous variables (age, height, weight, disease duration, T25FW, 9-hole peg test (9HPT), T2 lesion volume, Gad count, and NBV). Disease duration was estimated from the time of symptom onset.

**Table 4 | Estimated sample size for a one or two-year placebo-controlled randomized clinical trial of anti-CD2O-Abs, using different degrees of predictive enrichment**

| Percentile threshold[a] | CDP control[b] | CDP treatment[b] | HR (95% CI)[c] | Sample size estimate[d] | Number screened[e] |
|---|---|---|---|---|---|
| Two-year trial | | | | | |
| 0 | 0.30 | 0.24 | 0.74 (0.48–1.15) | 1374 | 1374 |
| 10 | 0.31 | 0.24 | 0.72 (0.46–1.13) | 1133 | 1259 |
| 20 | 0.30 | 0.22 | 0.70 (0.43–1.13) | 1019 | 1274 |
| 30 | 0.29 | 0.22 | 0.67 (0.40–1.12) | 812 | 1160 |
| 40 | 0.30 | 0.21 | 0.59 (0.33–1.03) | 464 | 773 |
| 50 | 0.33 | 0.20 | 0.49 (0.27–0.91) | 245 | 490 |
| 60 | 0.36 | 0.22 | 0.51 (0.26–0.98) | 251 | 628 |
| 70 | 0.39 | 0.19 | 0.36 (0.17–0.79) | 111 | 370 |
| One-year trial | | | | | |
| 0 | 0.20 | 0.12 | 0.74 (0.48–1.15) | 2435 | 2435 |
| 10 | 0.21 | 0.12 | 0.72 (0.46–1.13) | 1988 | 2209 |
| 20 | 0.20 | 0.11 | 0.70 (0.43–1.13) | 1796 | 2245 |
| 30 | 0.22 | 0.11 | 0.67 (0.40–1.12) | 1346 | 1923 |
| 40 | 0.25 | 0.11 | 0.59 (0.33–1.03) | 710 | 1183 |
| 50 | 0.26 | 0.11 | 0.49 (0.27–0.91) | 371 | 742 |
| 60 | 0.31 | 0.12 | 0.51 (0.26–0.98) | 365 | 913 |
| 70 | 0.30 | 0.10 | 0.36 (0.17–0.79) | 171 | 570 |

[a] Percentile threshold for randomization. The 0th percentile represents an unenriched population, while the 70th percentile leads to inclusion of only the top 30% who are predicted to be most responsive.

[b] Proportion of CDP24 events for the responder groups corresponding to each percentile threshold.

[c] HR for time-to-CDP24 for the responder groups corresponding to each percentile threshold.

[d] Sample size estimates are calculated using a desired power of 80% and $\alpha = 0.05$, assuming a 2:1 treatment to control randomization ratio. Calculations are based on the one or two-year CDP24 rate and one or two-year HR of responder groups in the anti-CD20-Abs dataset.

[e] Number of participants that need to be screened to reach the corresponding sample size estimate for randomization. This is dictated by the amount of predictive enrichment applied at randomization (see Percentile column).

Lesion segmentation and volumetric measurements are derived from ground-truth lesion masks, which were generated independently (by an image analysis center outside of this study) during the course of each clinical trial. A fully manual or a semi-automatic segmentation strategy was used during clinical trial analysis for each trial. This analysis began with automated segmentation and was followed by manual correction by experts. The resulting segmentation masks are the best available approximation to ground truth, but would not be expected to be identical between each expert and reading center in part due to differences in the approach to lesion segmentation between reading centers (school effects). To account for any difference between the trial sites' segmentation pipelines and improve model optimization dynamics[33], we scaled the segmentation-based metrics into a common reference range. To do so, we first isolated the subset of samples that fulfilled the intersection of inclusion criteria for all trials. Then, we scaled all MRI metrics such that their range from −3 SD to +3 SD matches that of a reference trial (in the same interval of ±3 SD) obtained from the training set. The reference trials were selected on the basis of sample size (ORATORIO for the PPMS trials, and OPERA I/II for the RRMS trials). The range was clamped at ±3 SD for the scaling to be robust to extreme outliers.

The following right-skewed distributions were log-transformed: NBV, T2 lesion volume, T25FW, and 9HPT. Gad counts were binned into bins of 0, 1, 2, 3, 4, 5-6, 7-9, 10-14, 15-19 and 20+ lesions. Finally, to improve convergence during gradient descent, all non-binary features were standardized by subtracting the mean and dividing by the standard deviation, both calculated from the training dataset[33].

**Outcome definition**

The primary outcome used in clinical trials assessing the efficacy of therapeutic agents on disease progression is the time to confirmed disability progression (CDP) at 12 or 24 weeks. We use CDP24 because it is a more robust indication that disability accrual will be maintained after 5 years[34]. CDP24 is most commonly based on the EDSS, a scale going from 0 (no disability) to 10 (death), in discrete 0.5 increments (except for a 1.0 increment between 0.0 and 1.0). A CDP24 event is defined as a 24-week sustained increase in the EDSS of 0.5 for baseline EDSS values > 5.5, of 1.5 for a baseline EDSS of 0, and of 1.0 for EDSS values in between. This difference in the increment required to confirm disability progression is commonly adopted in clinical trials, and partially accounts for the finding that patients transition through the EDSS scores at different rates[35].

While it is possible to predict time-to-event using traditional machine learning methods if workarounds are used to address right-censored data or using machine learning frameworks specifically developed to model survival data (reviewed elsewhere[36]), we chose not to model time-to-CDP24 because of limitations inherent in this metric. As outlined by Healy et al.[37], CDP reflects not only the rate of progression but also the baseline stage of the disease, which is problematic because the stage is represented by a discretized EDSS at a single baseline visit. This results in a noisy outcome label which could make it harder for a model to learn a representation that relates to the progressive biology which we are trying to model.

We therefore model the rate of progression directly by fitting a linear regression model onto the EDSS values of each individual participant over multiple visits (see Supplementary Methods 2 for details) and take its slope to be the outcome label that our MLP uses for training. One advantage of the slope outcome over time-to-CDP24 is that it can be modeled using any type of regression model. We revert to using time-to-CDP24 for model evaluation to facilitate comparison with treatment effect survival metrics reported in the original clinical trial publications.

**Treatment effect modeling**

To enrich clinical trials with individuals predicted to have an increased response to treatment, it is helpful to begin with the definition of

individual treatment effect (ITE) according to the Neyman/Rubin Potential Outcome Framework[38]. Let the ITE for individual $i$ be $\tau_i$, then

$$\tau_i := Y_i(1) - Y_i(0), \tag{1}$$

where $Y_i(1)$ and $Y_i(0)$ represent the outcome of individual $i$ when given treatment and control medications, respectively. The Fundamental Problem of Causal Inference[39] states that the ITE is unobservable because only one of the two outcomes is realized in any given patient, dictated by their treatment allocation. $Y_i(1)$ and $Y_i(0)$ are therefore termed potential outcomes or, alternatively, factual (observed) and counterfactual (not observed) outcomes.

Ground-truth can nevertheless be observed at the group level in specific situations, such as randomized control trials, because treatment allocation is independent of the outcome. We provide a detailed discussion of two important estimands, the average treatment effect (ATE) and the CATE in Supplementary Methods 1. Briefly, ATE represents the average effect when considering the entire population, while CATE considers a sub-population characterized by certain characteristics (e.g., 40-year-old women with 2 Gad lesions at baseline). We use CATE estimation to frame the problem of predicting treatment response for individuals.

The best estimator for CATE is conveniently also the best estimator for the ITE in terms of mean squared error (MSE)[6]. Several frameworks have been developed to model CATE, but a simple meta-learning approach which decomposes the estimation into sub-tasks that can be solved using any supervised machine learning model provides a flexible starting point[6]. For a broader survey of methods, see the survey on uplift modeling by Gutierrez et al.[4] (the uplift literature has contributed extensively to the field of causal inference, particularly when dealing with randomized experiments from an econometrics perspective).

In this work, an MLP was selected as the base model due to its high expressive power and flexibility to be integrated into larger end-to-end-trainable neural networks consisting of different modules (such as convolutional neural networks). We used a multi-headed architecture, with a common trunk and two output heads: one for modeling the potential outcome on treatment, $\hat{\mu}_1(x)$, and the other to model the potential outcome on placebo, $\hat{\mu}_0(x)$. For inference, the CATE estimate $\hat{\tau}(x)$ given a feature vector $x$ can be computed as:

$$\hat{\tau}(x) = \hat{\mu}_1(x) - \hat{\mu}_0(x). \tag{2}$$

We use $\hat{\tau}(x)$ as the predicted treatment effect for an individual with characteristics $x$. Note that we multiplied all $\hat{\tau}(x)$ values by −1 in this paper to simplify interpretation in the section "Results", such that a positive effect indicates improvement, while a negative effect indicates worsening on treatment.

This multi-headed approach can be seen as a variant of the T-Learner described for example by Kunzel et al.[6], except that the two base models in our case share weights in the common trunk. Our network is similar to that conceptualized by Alaa et al.[40], but without the propensity network used to correct for any conditional dependence between the treatment allocation and the outcome given the input features, since our dataset comes from randomized data.

To decrease the size of the hyperparameter search space, we fixed the number of layers and only tuned the layer width. We used one common hidden layer and one treatment-specific hidden layer. Additional common or treatment-specific layers could be used if necessary, but given the low dimensionality of our feature space and the relatively small sample size, the network's depth was kept small to avoid overfitting. The inductive bias behind our choice of using a multi-headed architecture is that disability progression can have both disease-specific and treatment-specific predictors of disability progression, which can be encoded into the common and treatment-specific hidden layer representations, respectively. Consequently, the common hidden layers can learn from all the available data, irrespective of treatment allocation. Rectified linear unit (ReLU) activation functions were used at hidden layers for non-linearity.

## Training
The model was trained in two phases, depicted in Fig. 3. In the first phase, a 5-headed MLP was pre-trained on an RRMS dataset to predict the slope outcome on each treatment arm. In the second phase, the parameters of the common layers were frozen, and the output heads were replaced with two new randomly initialized output heads for fine-tuning on the PPMS dataset to predict the same outcome.

Optimization was done using mini-batch gradient descent with momentum. To prevent overfitting, the validation loss was monitored during 4-fold cross-validation (CV) to early stop model training at the epoch with the lowest MSE, up to a maximum of 100 epochs. Dropout and L2 regularization were used, along with a max-norm constraint on the weights[41], to further prevent overfitting.

Mini-batches were sampled in a stratified fashion to preserve the proportions of participants receiving active treatment and placebo. Backpropagation was done using the MSE calculated at the output head that corresponds to the treatment that the patient was allocated to, $t_i$ (the output head with available ground-truth). The squared errors from each output head were then weighted by $n_s/(mn_t)$, where $n_s$ represents the total number of participants in the training split, $n_t$ represents the number of participants in the treatment arm corresponding to the output head of interest, and $m$ represents the total number of treatment arms. This compensates for the treatment allocation imbalance in the dataset.

We aimed to reduce variance by using the early stopped models obtained from each CV fold as members of an ensemble. This ensemble's prediction is the mean of its members' predictions, and is used for inference on the unseen test set.

A random search was used to identify the hyperparameters with the best validation performance (learning rate, momentum, L2 regularization coefficient, hidden layer width, max norm, dropout probability). We used CV aggregation, or crogging[42], to improve the generalization error estimate using our validation metrics. Crogging involves aggregating all validation set predictions (rather than the validation metrics) and computing one validation metric for the entire CV procedure. The best model during hyperparameter tuning was selected during CV on the basis of two validation metrics: the MSE of the factual predictions, and the $AD_{wabc}$ (described in detail in Supplementary Methods 3). We combine both validation metrics during hyperparameter tuning by choosing the model with the highest $AD_{wabc}$ among all models that fall within 1 SD of the best performing model based on the MSE loss. The SD of the best performing model's MSE is calculated from the loss values obtained in the individual CV folds.

## Baseline models
The performance of the multi-headed MLP was compared to ridge regression and CPH models. Both models were used as part of a T-learner configuration (as defined by Kunzel et al.[6]). Hyperparameter tuning was done on the same folds and with the same metrics as for the MLP.

## Statistical analysis
Hazard ratios were calculated using CPH models and associated p-values from log-rank tests. Sample size estimation for CPH assumes a two-sided test and was based on Rosner[43], as implemented by the Lifelines library (version 0.27.0)[44].

## Software
All experiments were implemented in Python 3.8[45]. MLPs were implemented using the Pytorch library (version 1.7.1)[46]. Scikit-Learn (version

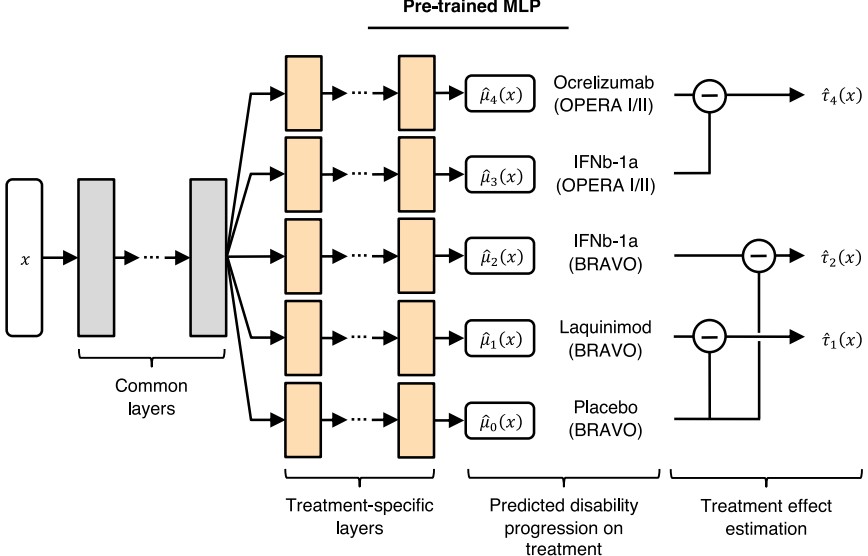

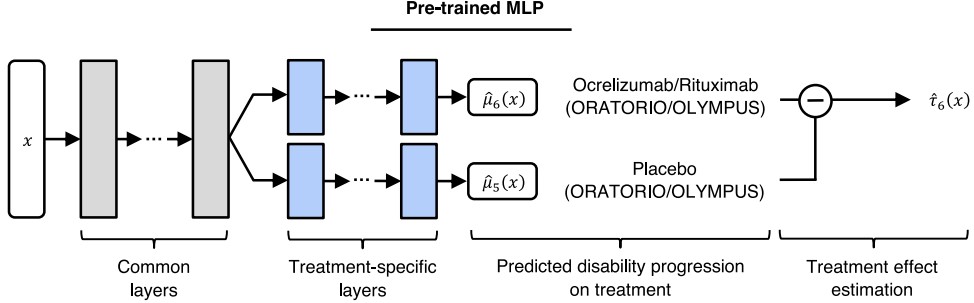

**Fig. 3 | Multi-headed multilayer perceptron (MLP) architecture for CATE esti-mation.** The MLP was first pre-trained on a relapsing-remitting multiple sclerosis dataset (top), followed by fine tuning on a primary progressive multiple sclerosis dataset (bottom). Subtraction symbols indicate which treatment and control are being subtracted for the CATE estimate. Gray-colored layers indicate the common layers that are transferred from the pre-trained MLP to the fine-tuning MLP, at which point their parameters are frozen and only the parameters of the blue-colored layers are updated. The orange-colored layers are discarded after the pre-training step. $x$: Feature vector. $\hat{\tau}_t(x)$: CATE estimate for treatment $t$ given feature vector $x$. $\hat{\mu}_t(x)$: predicted potential outcome on treatment $t$. IFNb-1a = Interferon beta-1a.

0.24.2)[47] was used for the implementation of ridge regression, while Lifelines (version 0.27.0)[44] was used for CPH. For reproducibility, the same random seed was used for data splitting and model initialization across all experiments.

### Reporting summary
Further information on research design is available in the Nature Research Reporting Summary linked to this article.

## Data availability
Data used in this work was obtained from the following clinical trials (OPERA I[8], OPERA II[8], BRAVO[9], ORATORIO[10], OLYMPUS[11], and ARPEGGIO[12] with ClinicalTrials.gov numbers NCT01247324, NCT01412333, NCT00605215, NCT01194570, NCT00087529, NCT02284568, respectively), and are not publicly available. Access requests should be forwarded to the relevant data controllers.

## Code availability
Code necessary to reproduce the proposed methodological frame-work can be accessed publicly on the following GitHub repository: https://github.com/jpfalet/ms-predictive-enrichment. This repository does not contain dataset-specific code, since the data we used is not publicly available.

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

## Acknowledgements
The authors are grateful to the companies who generously provided the clinical trial data that made this work possible: Biogen, BioMS, MedDay, Novartis, Roche / Genentech, and Teva.

This investigation was supported by an award from the International Progressive Multiple Sclerosis Alliance (award reference number PA-1412-02420), the Natural Sciences and Engineering Research Council of Canada (RGPIN-2015-05471, T.A.), the Canada Institute for Advanced Research (CIFAR) Artificial Intelligence Chairs program (T.A.), a technology maturation grant from Mila—Quebec AI Institute (T.A.), an endMS Personnel Award from the Multiple Sclerosis Society of Canada (J.-P.R.F.), a Canada Graduate Scholarship-Masters Award from the Canadian Institutes of Health Research (J.-P.R.F.), and the Fonds de recherche du Québec—Santé/Ministère de la Santé et des Services sociaux training program for specialty medicine residents with an interest in pursuing a research career, Phase 1 (J.-P.R.F.).

## Author contributions
J.-P.R.F., T.A., and D.L.A. designed the study. J.-P.R.F., J.D.-F., B.N., J.S., F.B., M.-P.S., D.P., T.A., and D.L.A. contributed to the methods, analysis, and writing the manuscript. J.-P.R.F. wrote the code for all experiments. T.A. and D.L.A. jointly supervised this work. D.L.A. oversaw data collection.

## Competing interests
F.B. has received teaching honoraria from Novartis and has received personal compensation for consulting services from Biogen, Eisai and Chiesi. M.-P.S., has received personal compensation for consulting services and for speaking activities from Merck, Teva, Novartis, Roche, Sanofi Genzyme, Medday, GeNeuro, and Biogen. D.P. works part-time for DeepMind. D.L.A. reports consulting fees from Biogen, Celgene, Frequency Therapeutics, Genentech, Merck, Novartis, Race to Erase MS, Roche, and Sanofi-Aventis, Shionogi, Xfacto Communications, grants from Immunotec and Novartis, and an equity interest in NeuroRx. The remaining authors report no competing interests.
