## [Peer Review File · Nature Communications]

Estimating individual treatment effect on disability progression in multiple sclerosis using deep learningEditorial Note: This manuscript has been previously reviewed at another journal that is not operating a transparent peer review scheme. This document only contains reviewer comments and rebuttal letters for versions considered at *Nature Communications*.

REVIEWERS' COMMENTS

Reviewer #1 (Remarks to the Author):

The manuscript entitled "Estimating treatment effect for individuals with progressive multiple sclerosis using deep learning" has undergone major revisions, and some of the prior points have been addressed. However, several major issues remain, and some contradictions in predicting treatment response with the literature are introduced that are difficult to understand. I have elaborated below and hope that this will help the manuscript to improve even further:

The study is not about progressive MS. Instead, revisions have added RRMS patients (<800 individuals). While this is commendable, the models are trained and assessed on a majority RRMS population (understandably so given that RRMS is much more common, but this contradicts the claims on finding specific progressive MS models). In phase one of training, all patients are RRMS. Additionally, the model performs much better for patients with disability levels that are usually not considered progressive MS (EDSS<4.5). Thirdly, the most important factor (if I understand Table 3 correctly) is the T2 lesion volume. For these reasons, I suggest changing the title and the focus throughout from PMS to MS in general. This, however, brings a question on the novelty of this study in understanding mechanisms of action given the number of prior studies (Dr Rio's treatment response scores, and Dr Sormani's prior work, and several other studies) on RRMS that have identified similar factors and many more on disease activity using transparent (as opposed to black-box) methods.

Ditto, as suggested by other reviewers, I suggest removing claims on finding specific mechanisms and revising those claims to find individuals' characteristics that contribute to a more beneficial treatment response in light of the new analyses. This issue has been overlooked.

The revision has brought forward one major issue. There is a lack of uniform image processing, which was not clear in the previous version but adding a standardization has changed almost all the important factors related to treatment response. I sympathize with the fact that this important study brings together older trials that have had different scanners and acquired at different times; however, a minimum expectation is to use a uniform image processing pipeline to reduce confounders and focus on biological questions. This had caused a significant discrepancy in the pre-revision version, in which disease activity differed in the training and test set. However, in the revision, the expectation is to use a known and validated harmonization method based on image processing techniques rather than simply normalizing values, which will not address the fundamental issue of trial confound (that is, using different image processing techniques in trials that have been acquired possibly decades away from each other). The scaling factor calculated and used is not validated before. I suggest redoing this part by processing MRI scans using a uniform tool for all trials. The response to the review brings this forward and is unsatisfactory: "ARPEGGIO used a different segmentation algorithm with a different sensitivity for lesions, and the number/size of lesions identified was smaller than the other trials. The model, therefore, was unlikely to generalize what it had learned about the lesion metrics to this dataset. We have now corrected this by scaling all the segmentation-related metrics to a common range (described in the Methods)."

The revision has not fully addressed the issue of sample size in progressive MS. The title refers to predicting treatment effect in progressive MS, yet new revisions have added 803 subjects to the relapsing-remitting MS population. My comment and that of editors in prior revision have not been addressed. Changing the title and editing the manuscript, however, should address this.

Presented final results on responder vs non-responder in figure 3 belong to the training cohort. The figure should change and show only the external validation cohorts (ORATORIO, etc.). It is unclear

from the text as it stands how many ORATORIO patients have been identified in this responder group. From supplementary figure one, it looks as if this group has less than 30 patients. The issue of sample size in the progressive MS trials is therefore not addressed. The responder group in laquinimod (I assume this is Arpeggio trial) is again small (14 vs five by the end of year 2), and the confidence intervals are very wide (supplemental figure 2) to provide reliable estimates.

It is odd that in a clinical trial with proven treatment effects (ocrelizumab), the hazard ratio is 0.36, but in a clinical trial of a failed treatment (laquinimod), the HR of 30% of the most likely responders is 0.28.

It is unexpected that in the ORATORIO trial for the 50th percentile of responders, the HR of disability progression is not significant. In the original publication (Montalban et al., 2018, New England Journal of Medicine), the HR for the whole group was significant. Therefore, I would expect that half of those who are predicted by authors' model to be most likely to respond, should also have a significant HR. It is hard to convince the reader that the presented results are reliable and in line with the literature. Computer code and statement on their availability should be added. This is important for reproducibility.

Reviewer #2 (Remarks to the Author):

Comments for "Estimating treatment effect for individuals with progressive multiple sclerosis using deep learning"

The authors have addressed my comments on the previous version of the manuscript. I have a couple of additional comments.

Comment:

1) Page 5/ Table 3: Is there a way to compare the values for the ADCwabc across the models? I understand the rank ordering of the values, but I do not have much intuition regarding the magnitude of the differences between the models. This would help to understand whether the models or subgroups with higher values are slightly or much different from the models or subgroups with lower values.

2) Page 5: I believe the p-value for the first hazard ratio in Section 2.3 is incorrect. Also, should the ADCwabc in Section 2.3 match the value from Table 3 (0.0208 vs 0.0211)?

3) Page 6: You state that the prognostic MLP model is second best in the Anti-CD20-Abs dataset, but it seems that it is fourth after the two other versions of MLP and T2 lesion volume/ disease duration.

4) Page 6: You state, "indicating no improvement in treatment effect compared to the whole group" for both Anti-CD20-Abs and laquinimod, but the HR for the laquinimod subgroup is smaller than either of the HRs you identified in your predictive enrichment groups from section 2.3 (0.305 vs. 0.492 and 0.338).

5) Page 6: Could you provide more details regarding your sample size calculation? I tried to reproduce your calculation and I got similar sample size calculations, but I would like to know the exact approach so it could be reproduced.

6) Page 13: Was the ridge regression model fit with the EDSS slope outcome?

7) Table 1: Could you check the RMST? It seems too close to 2 since it is calculated at 24 months.

Reviewer #3 (Remarks to the Author):

The authors have carefully considered the feedback from the reviewers, and have amended the manuscript as appropriate.

I have no further comments.

Reviewer #1 (Remarks to the Author):

The manuscript entitled "Estimating treatment effect for individuals with progressive multiple sclerosis using deep learning" has undergone major revisions, and some of the prior points have been addressed. However, several major issues remain, and some contradictions in predicting treatment response with the literature are introduced that are difficult to understand. I have elaborated below and hope that this will help the manuscript to improve even further:

The study is not about progressive MS. Instead, revisions have added RRMS patients (<800 individuals). While this is commendable, the models are trained and assessed on a majority RRMS population (understandably so given that RRMS is much more common, but this contradicts the claims on finding specific progressive MS models). In phase one of training, all patients are RRMS. Additionally, the model performs much better for patients with disability levels that are usually not considered progressive MS (EDSS<4.5). Thirdly, the most important factor (if I understand Table 3 correctly) is the T2 lesion volume. For these reasons, I suggest changing the title and the focus throughout from PMS to MS in general. This, however, brings a question on the novelty of this study in understanding mechanisms of action given the number of prior studies (Dr Rio's treatment response scores, and Dr Sormani's prior work, and several other studies) on RRMS that have identified similar factors and many more on disease activity using transparent (as opposed to black-box) methods.

Response: We thank the reviewer for their comment. We would like to apologize if some of the comments we made in our manuscript lacked clarity. First, we have added 2,520 new RRMS patients to our training dataset, and not < 800. This is stated in Section 2.1, second paragraph. Second, the RRMS dataset is used only for pre-training, and the model is then fine-tuned and tested solely on the progressive MS subset, which is the main focus of our work. This "transfer learning" strategy was implemented to mitigate the issue of small sample size in progressive MS patient datasets, a point previously brought up by one reviewer. Pre-training on a related dataset is a common technique in the field of machine learning to increase the robustness of learned representations when dealing with smaller sample sizes. This was already explained in section 2.1, second paragraph. Third, while it is true that secondary progressive MS patients usually have higher EDSS scores, this is not the case with primary progressive MS patients which begin progression from the onset of their disease (EDSS can range from 0 to 10). Since our evaluation dataset includes patients with primary progressive MS, the fact that our model performs better in those with EDSS < 4.5 is not problematic.

However, we appreciate that the work involved both RRMS and progressive MS, and that similar pathophysiological processes occur in both clinical subtypes. We therefore agree to change the title and focus throughout the paper to be about progression in MS in general. We have also clarified which sub-type of MS is used for which phase of our analysis in the abstract and main text (Section 2.1).

Ditto, as suggested by other reviewers, I suggest removing claims on finding specific mechanisms and revising those claims to find individuals' characteristics that contribute to a more beneficial treatment response in light of the new analyses. This issue has been overlooked.

Response: We make no claim about finding specific mechanisms in our manuscript. The other reviewers do not suggest removing any specific mechanistic claim from this version of the manuscript (please refer to their comments below). In the absence of a more specific request, we are therefore under the impression that no further changes are needed.

The revision has brought forward one major issue. There is a lack of uniform image processing, which was not clear in the previous version but adding a standardization has changed almost all the important factors related to treatment response. I sympathize with the fact that this important study brings together older trials that have had different scanners and acquired at different times; however, a minimum expectation is to use a uniform image processing pipeline to reduce confounders and focus on biological questions. This had caused a significant discrepancy in the pre-revision version, in which disease activity differed in the training and test set. However, in the revision, the expectation is to use a known and validated harmonization method based on image processing techniques rather than simply normalizing values, which will not address the fundamental issue of trial confound (that is, using different image processing techniques in trials that have been acquired possibly decades away from each other). The scaling factor calculated and used is not validated before. I suggest redoing this part by processing MRI scans using a uniform tool for all trials. The response to the review brings this forward and is unsatisfactory: "ARPEGGIO used a different segmentation algorithm with a different sensitivity for lesions, and the number/size of lesions identified was smaller than the other trials. The model, therefore, was unlikely to generalize what it had learned about the lesion metrics to this dataset. We have now corrected this by scaling all the segmentation-related metrics to a common range (described in the Methods)."

Response: We appreciate the reviewer's comment on harmonization, which made us realize that our previous response to this question was unclear and could have led to confusion. The segmentation-based metrics, particularly the T2 lesion volumes, used in this study are derived from the ground-truth lesion masks, which were generated independently (by an image analysis centre outside of this study) during the course of each clinical trial.

A fully manual or a semi-automatic segmentation strategy was used during clinical trial analysis for each trial. This analysis began with automated segmentation and was followed by manual correction by experts. The automated segmentation algorithms were proprietary for the image analysis centre performing the measurements and are not available to external investigators (such as ourselves). In addition, there are well known differences in the approach to determining lesion boundaries (school-effects) that can result in substantial differences in lesion volumes between different reading centers. Thus, the lesion masks we used are the best approximation we have to ground truth, but would not be expected to be identical between each expert and reading centre.

Repeating this process using a consistent but different image processing pipeline and different expert annotators would not necessarily result in "better" segmentation masks, and is impractical as this type of work is extremely labour intensive and originally cost millions of dollars to perform. It is highly unlikely that any other investigator would have the resources to reproduce our work or adapt it to new data if we were to do this. The approach we have taken is much more practical in that it standardizes the range of input data to account for these school-effects, and therefore only requires access to the lesion counts/volumes that were generated during a clinical trial, thus making it feasible for other investigators to use this approach on other datasets obtained under different circumstances.

In fact, scaling the range (and/or shifting the mean) of a feature's input distribution to a reference obtained from a training dataset is common practice in machine learning and serves to improve model optimization dynamics [Lecun et al. Efficient Backprop. *Neural Networks: Tricks of the Trade* 9–48 (2012)]. All our input features (not just the segmentation-based metrics) are normalized (or in some cases log-transformed) precisely for this reason. This was already explained in the last paragraph of section 4.1, but we have expanded on our rationale in section 4.1, third paragraph.

Finally, we analysed data from ARPEGGIO independently from the other trials, so our conclusions about the characteristics of responders versus non-responders in ARPEGGIO should not be significantly affected by the aforementioned school-effects.

In addition, please note that we are not the only group to have also published treatment effect analyses using ground-truth data from a multi-centre, federated dataset [see Bovis et al. *BMC Med.* 2019 Jun 18;17(1):113.].

The revision has not fully addressed the issue of sample size in progressive MS. The title refers to predicting treatment effect in progressive MS, yet new revisions have added 803 subjects to the relapsing-remitting MS population. My comment and that of editors in prior revision have not been addressed. Changing the title and editing the manuscript, however, should address this.

Response: Once again we apologize if this was not clear. We have added 2,520 RRMS patients, and not 803. The title and focus of the manuscript has been made more general, as suggested.

Presented final results on responder vs non-responder in figure 3 belong to the training cohort. The figure should change and show only the external validation cohorts (ORATORIO, etc.).

Response: We apologize if this was also unclear. All figures and tables reporting evaluation metrics concern the external validation cohorts and not the training cohort. This was previously explained in our results and methods sections, but we have now further emphasized the testing cohort in each figure legend and table caption, as well as in the main text (section 2.2, second and third paragraphs).

It is unclear from the text as it stands how many ORATORIO patients have been identified in this responder group. From supplementary figure one, it looks as if this group has less than 30 patients. The issue of sample size in the progressive MS trials is therefore not addressed. The responder group in laquinimod (I assume this is Arpeggio trial) is again small (14 vs five by the end of year 2), and the confidence intervals are very wide (supplemental figure 2) to provide reliable estimates.

Response: We appreciate the reviewer's comment on sample size, which was shared by other reviewers in the previous review, and which we had therefore addressed in our previous response. The primary aim of our work is to address the scarcity of efficacious disease modifying treatments in primary progressive MS, for which only one drug (ocrelizumab) has been found to slow disability progression in phase 3 clinical trials. We already include data from the phase 3 clinical trial that studied ocrelizumab (ORATORIO) as well as data from another trial studying a drug with a similar mechanism of action (OLYMPUS) as part of our external validation set, and we further enhanced our validation set by including a third trial (ARPEGGIO). Ultimately, the size of this external validation set is limited by what is currently available.

That said, the size of the responder groups is larger than what is quoted by the reviewer, and all sample sizes are mentioned in Table 2, Figure 2, and Supplementary Figure 2. Even in the most restrictive case, the smallest responder group we include in our main analysis is of size 92, and this is when looking at the top 30% most responsive individuals to anti-CD20-abs, which is more restrictive than other response thresholds we report on (such as the top 50% responders, which has a sample size of 151 individuals). For responders to laquinimod, the sample sizes are 99 for the top 30% and 159 for the top 50%.

The reviewer is referring specifically to the sample size for the responders we identified in a sub-group analysis where we only looked at ORATORIO participants (a sub-group analysis which was requested by a reviewer in the previous review), which is of course smaller than the sample sizes in our main analysis. However, this still amounts to 57 individuals, and not < 30. To make these sub-group numbers clear to the reader, we have added them to the main text (section 2.2, third paragraph).

It is odd that in a clinical trial with proven treatment effects (ocrelizumab), the hazard ratio is 0.36, but in a clinical trial of a failed treatment (laquinimod), the HR of 30% of the most likely responders is 0.28.

Response: Unfortunately, the numbers quoted by the reviewer do not align with what is in the text. The stated HR of 0.36 is not for the responder group of ocrelizumab patients, but rather for the responder group for the ocrelizumab + rituximab group (section 2.2, paragraph 3). The stated HR of 0.28 is not for the laquinimod group, but for the ocrelizumab group (section 2.2, paragraph 4). The correct HR for the laquinimod group is 0.338 (section 2.3, paragraph 1). There is therefore no inconsistency with the previously published effect sizes for these two medications.

It is unexpected that in the ORATORIO trial for the 50th percentile of responders, the HR of disability progression is not significant. In the original publication (Montalban et al., 2018, New England Journal of Medicine), the HR for the whole group was significant. Therefore, I would expect that half of those who are predicted by authors' model to be amongst likely to respond, should also have a significant HR. It is hard to convince the reader that the presented results are reliable and in line with the literature.

Response: We do not base our conclusions on this sub-group analysis of the single trial ORATORIO, which was requested by a previous reviewer. Our conclusions are based on the larger sample size of the combined ORATORIO and OLYMPUS dataset, which both studied anti-CD20 monoclonal antibodies. Nonetheless, our subgroup analysis is still in line with our findings on the combined dataset, and show a gradual increase in the effect size as we select individuals that are predicted to be more responsive. The fact that a single threshold for including responders (50% in this example) does not reach statistical significance in a subgroup analysis which effectively boils down to 15% of the total ORATORIO clinical trial dataset is not unexpected and would not change our main conclusions.

Computer code and statement on their availability should be added. This is important for reproducibility.

Response: We have made our code publicly available on GitHub and have added a statement to this effect in the paper.

Reviewer #2 (Remarks to the Author):

Comments for "Estimating treatment effect for individuals with progressive multiple sclerosis using deep learning"

The authors have addressed my comments on the previous version of the manuscript. I have a couple of additional comments.

Comment:

1) Page 5/ Table 3: Is there a way to compare the values for the ADCwabc across the models? I understand the rank

ordering of the values, but I do not have much intuition regarding the magnitude of the differences between the models. This would help to understand whether the models or subgroups with higher values are slightly or much different from the models or subgroups with lower values.

Response: We have added Supplementary Figure 3 and 4, which shows the model performance in a scatter plot to facilitate this type of comparison.

2) Page 5: I believe the p-value for the first hazard ratio in Section 2.3 is incorrect. Also, should the ADCwabc in Section 2.3 match the value from Table 3 (0.0208 vs 0.0211)?

Response: Indeed, there was a mistake with the first p value of Section 2.3, which should be 0.176. This is now corrected. The stated discrepancy between the two ADwabc values was also a mistake, and the two now show the correct value of 0.0211.

3) Page 6: You state that the prognostic MLP model is second best in the Anti-CD20-Abs dataset, but it seems that it is fourth after the two other versions of MLP and T2 lesion volume/ disease duration.

Response: This was corrected in section 2.4, paragraph 1, last sentence.

4) Page 6: You state, "indicating no improvement in treatment effect compared to the whole group" for both Anti-CD20-Abs and laquinimod, but the HR for the laquinimod subgroup is smaller than either of the HRs you identified in your predictive enrichment groups from section 2.3 (0.305 vs. 0.492 and 0.338).

Response: We thank the reviewer for noticing this error, and have replaced this sentence with the correct interpretation.

5) Page 6: Could you provide more details regarding your sample size calculation? I tried to reproduce your calculation and I got similar sample size calculations, but I would like to know the exact approach so it could be reproduced.

Response: Details were added in the section 4.6 (Statistical Analysis).

6) Page 13: Was the ridge regression model fit with the EDSS slope outcome?

Response: All baseline comparator models (including the ridge regression model) were fit using the same EDSS slope outcome. This was clarified in the second sentence of the first paragraph in Section 2.4.

7) Table 1: Could you check the RMST? It seems too close to 2 since it is calculated at 24 months.

We verified this finding, and it is correct. At 2 years, as shown in the first row of Table 4, the proportion of CDP is relatively small, which is why the RMST (which reflects the area under the Kaplan-Meier curve, clamped at 2 years) is close to 2 years.

Reviewer #3 (Remarks to the Author):

The authors have carefully considered the feedback from the reviewers, and have amended the manuscript as appropriate.

I have no further comments.